# Heart Rate Variability in Subjects with Severe Allergic Background Undergoing COVID-19 Vaccination

**DOI:** 10.3390/vaccines11030567

**Published:** 2023-03-01

**Authors:** Maria Bernadette Cilona, Filippo D’Amico, Chiara Asperti, Giuseppe Alvise Ramirez, Stefano Turi, Giovanni Benanti, Shai Marc Bohane, Serena Nannipieri, Rosa Labanca, Matteo Gervasini, Federica Russetti, Naomi Viapiana, Martina Lezzi, Giovanni Landoni, Lorenzo Dagna, Mona-Rita Yacoub

**Affiliations:** 1Unit of Immunology, Rheumatology, Allergy and Rare Diseases, IRCCS San Raffaele Scientific Institute, IRCSS San Raffaele Hospital, Via Olgettina, 60, 21132 Milan, Italy; 2Department of Anesthesia and Intensive Care, IRCCS San Raffaele Scientific Institute, 21132 Milan, Italy; 3School of Medicine, Vita-Salute San Raffaele University, Via Olgettina, 60, 21132 Milan, Italy

**Keywords:** heart rate variability, COVID-19, vaccination, hypersensitivity reactions, autonomic nervous system, SARS-CoV-2

## Abstract

Anti-Severe Acute Respiratory Syndrome Coronavirus 2 (SARS-CoV-2) vaccination is the world’s most important strategy for stopping the pandemic. Vaccination challenges the body’s immune response and can be complicated by hypersensitivity reactions. The autonomic nervous system can modulate the inflammatory immune response, therefore constituting a potential marker to characterize individuals at high risk of hypersensitivity reactions. Autonomic nervous system functionality was assessed through measurement of the heart rate variability (HRV) in subjects with a history of severe allergic reactions and 12 control subjects. HRV parameters included the mean electrocardiograph RR interval and the standard deviation of all normal R–R intervals (SDNN). All measurements were performed immediately before the anti-SARS-CoV-2 vaccination. The median RR variability was lower in the study than in the control group: 687 ms (645–759) vs. 821 ms (759–902); *p* = 0.02. The SDNN was lower in the study group than in the control group: 32 ms (23–36) vs. 50 ms (43–55); *p* < 0.01. No correlation was found between age and the SDNN. Autonomic nervous system activity is unbalanced in people with a severe allergy background.

## 1. Introduction

Coronavirus disease 2019 (COVID-19) is a highly transmissible respiratory infection caused by the severe acute respiratory syndrome coronavirus 2 (SARS-CoV-2) and is characterized by a respiratory syndrome and a heterogeneous hyperinflammatory response [1]. Global healthcare systems have faced an unprecedented challenge. Despite improvements in the medical management of COVID-19, vaccination against the viral pathogen SARS-CoV-2 represents the most important global strategy in controlling the pandemic. Indeed, vaccination campaigns rapidly advanced, especially in developed countries, and are associated with a clear reduction in the number of SARS-CoV-2 infections, decrease in viral RNA load, reduction in illness duration and attenuation of symptoms among those with breakthrough infections despite vaccination [2]. However, vaccination campaigns encountered some hurdles. People’s adherence to large-scale vaccination was limited in part due to their fear of any adverse reactions to vaccination [3]. The side effects of vaccination were amplified by the media, increasing vaccine hesitancy from the beginning of the vaccination campaign, and governments introduced measures to undermine misinformation and encourage vaccination. Scientific research has currently been focused on hypersensitivity reactions and the characteristics of subjects who may develop them [4,5]. Hypersensitivity reactions are an exaggerated and inappropriate immunological response to an allergen typically characterized by skin, respiratory, and/or cardiovascular involvement [6]. Individuals who have experienced a suspected hypersensitivity reaction to the vaccine or to its excipient usually undergo a preventive allergy workup to safely receive another dose of the vaccine, which includes allergy tests with vaccine excipients (polyethylene glycol and polysorbates) in selected cases, as suggested by national and international guidelines [7].

The pathophysiological mechanisms underlying hypersensitivity reactions involve mast cells’ degranulation, Th2 response, and alteration of the neuroendocrine–immune axis [8]. Notably, the autonomic nervous system plays a key role in all those mechanisms involved in the activation of inflammatory immune responses [8,9]. Indeed, allergic reactions can occur at different levels of organ involvement (e.g., respiratory, gastrointestinal, cutaneous) and may have an impact on hemodynamic stability. In acute conditions such as anaphylactic reactions, a sudden drop in blood pressure is caused by vasodilation induced by mast cells [10]. Gastrointestinal manifestations are characterized by increased peristalsis, mucus production, and diarrhea [11]. Typically, atopic dermatitis is characterized by a chronic itch caused by increased histamine production and inflammatory skin lesions [12]. Chronic conditions such as allergic rhinitis and asthma are mediated by mucus production and bronchial overactivity which result in bronchoconstriction [13]. The nervous system is involved in regulating all these pathophysiological mechanisms. Furthermore, due to the deep innervation of visceral organs, neurons share proximity with immune cells, leading to a peculiar crosstalk mediating the inflammatory immune response. In particular, the sympathetic nervous system leads to dysregulated cytokine production and unbalanced Th1/Th2 responses, stimulates mast cell degranulation, and activates inflammatory mediators and hormones directly involved in inflammation (e.g., cortisol, histamine, vasoactive intestinal peptide) [14]. Indeed, the finding of a sympathetic nervous system activation could represent a marker of the degree of inflammatory response and a possible therapeutic target of allergic conditions [15]; however, a useful marker of the sympathetic nervous system activity is heart rate variability (HRV). The HRV describes the oscillation of the intervals between consecutive heartbeats (the time elapsed between two successive R waves of the QRS signal (R–R intervals)) and reflects cardiac autonomic modulation [16]. In physiologic conditions, heart rate and rhythm are regulated by the intrinsic cardiac system of specialized pace-maker cells and modulated by the autonomic nervous system. The activation of the sympathetic nervous system tends to synchronize the RR interval to decrease the HRV. A widely used measure is the standard deviation of all normal RR intervals (SDNN) [17]. The HRV can be used in clinical practice to assess the physiologic status of autonomic activity, and HRV measurement is routinely used in cardiovascular medicine and sports medicine [18,19]. The HRV is often used for professional athletes to monitor cardiac activity. Athletes use these parameters to measure their psychophysical balance and response to physical stress [20]. Despite HRV being applied in clinical practice for a wide spectrum of diseases, consensus was only reached to use HRV parameters to predict the risk of arrhythmic events after acute myocardial infarction and function as a clinical marker for progression of diabetic neuropathy [21]. In all the other medical fields, there are insufficient clinical data on the practical utility of HRV. Indeed, further studies are needed to confirm its potential utility in clinical practice in other areas. In the field of allergology, the HRV was used to characterize subjects with chronic immunological disease [22,23]. The role of the autonomic nervous system has never been studied in acute conditions such as hypersensitivity reactions.

This study aims to compare the HRV in subjects with a severe allergic background undergoing anti-SARS-CoV-2 vaccination with the HRV of subjects from a control group. 

## 2. Materials and Methods

### 2.1. Study Design

We conducted an observational 2:1 case–control study comparing individuals with history of severe allergic reactions undergoing anti-SARS-CoV-2 vaccination in a dedicated prompt-response setting with individuals vaccinated in a standard environment. Participants were previously identified by an allergist not involved in study data collection and defined as people with a history of multiple or severe anaphylaxis, severe controlled asthma, chronic spontaneous urticaria with a recent flare, mast cells disorders, or previous hypersensitivity reactions to SARS-CoV-2 vaccines. All participants were enrolled between February and March 2022, and vaccination sessions were conducted in IRCCS San Raffaele Hospital, Milan. The primary endpoint was the difference in the SDNN between the study and control groups. The secondary endpoints were correlations between age and SDNN and the difference in RR values between the study and control groups.

### 2.2. Enrollment

Participants were enrolled, upon signing a specific informed consent form, in the Panimmuno research protocol approved by the Institutional Review Board (reference code 22/INT/2018). Subjects undergoing vaccination and older than 18 years were eligible for the study. Patients with cardiac arrhythmias, taking rate/rhythm control drugs, or who did not provide written informed consent to participate in the study were excluded. Subjects with a severe allergic background were previously identified by an allergist independently from the study. Those individuals considered to have a high risk of hypersensitivity reactions were referred to exclusive vaccination sessions. Subjects with a history of severe hypersensitivity reactions were identified by anamnestic data through the administration of a questionnaire assessing their history of atopy, antiallergic medication, previous allergic reactions to drugs and/or food, or reactions to previous doses of an COVID-19 vaccine. The control group was selected according to the expected standards for this type of study. Indeed, the control group was defined as a group of subjects undergoing COVID-19 vaccination with a low risk of developing hypersensitivity reactions to vaccination. The low risk of developing hypersensitivity reactions to vaccination was defined by the allergist based on the subject’s clinical history. Due to the limited number of subjects undergoing vaccination in the later stages of vaccination campaign, subjects were sequentially selected; therefore, we decided to select the first 24 subjects of the study group and the first 12 subjects of the control group meeting the inclusion criteria for their enrolment in the study. 

### 2.3. Sample Size

Available data in the literature have suggested a normal value of the SDNN of 50 ± 10 ms [16].

The SDNN < 50 ms is associated with an overactivity of the sympathetic nervous system. In the latest meta-analysis on the association of inflammation and the HRV, they found the SNDD was the strongest value negatively correlated with markers of inflammation [24]. Previous studies have reported values close to 40 ms are associated with clinical manifestations that typically result from an overactivation of the sympathetic nervous system [25,26,27,28].

Therefore, we identified a reduction in the SDNN to 40 ms for the present study to be significantly relevant. A sample size calculation based on Pearson’s Chi-square test with a two-sided alpha error of 0.05 and 80% power with an enrolled ratio of 2 suggested a sample size of 24 participants in the study group and 12 participants in the control group using the continuity correction, resulting in a total study population of 36 patients. 

### 2.4. Outcome Measurement

On the day of injection, participants were monitored for 5 min in an isolated room before vaccine administration, and their mean RR variability and SDNN were evaluated. Data were analyzed and elaborated with the previously validated Kubios HRV software (version 3.5; University of Kuopio, Kuopio, Finland) [29], according to the guidelines recommended by the Taskforce of the European Society of Cardiology and the North American Society of Pacing and Electrophysiology [16]. An example of data extraction is reported in Appendix A). As a secondary analysis, we correlated the age of individuals in the study group with SDNN values. All patients were also screened by means of an allergy questionnaire reported in Appendix A. The assessed characteristics included the following: history of vaccination, hypersensitivity reactions to vaccination, allergic drug reactions, positivity to allergy tests for common aeroallergens, other allergic comorbidities, and antiallergic drugs consumed. Demographic data, clinical history, and medications were also collected on the day of vaccination. 

### 2.5. Statistical Analysis

Demographic and baseline disease characteristics were summarized with the use of descriptive statistics. Categorical variables were reported as absolute numbers and percentages. Unadjusted univariate analyses to compare the two treatment groups were based on Fisher’s exact test. Continuous variables were reported as median, interquartile range (IQR). Normality was evaluated using the Shapiro–Wilk test. Between-group differences were evaluated using the *t*-test or Wilcoxon signed-rank test in accordance with normality of the distribution [30]. We considered significant a *p*-value < 0.05. Correlation was calculated using a Pearson correlation test, considering an inconsistent correlation for R < 0.1, weak correlation for R < 0.5, and strong correlation for R > 0.5 [31]. Subjects of the study group had peculiar characteristics related to their allergic history (e.g., antiallergic medications, positivity to allergic test); therefore, it was not possible to match individuals of the study group with those of the control group. Considering the small sample size and the limitations in enrolling more subjects, we decided not to adjust the baseline characteristics between the groups to avoid errors in the data interpretation [32].

## 3. Results

### 3.1. Study Population

Of the 50 screened subjects, 36 were eligible for the study, and the main reason for exclusion was based on medications able to influence a subject’s heart rate (Figure 1). As planned, we enrolled 24 patients in the study group and 12 in the control group. Individuals in the study group were younger (48 years (34–53) vs. 60 years (53–67); *p* < 0.01), and more frequently female (96% vs. 75%, *p* = 0.02) than those in the control group (Table 1). Individuals in the study group also more frequently had a history of hypersensitivity reactions to drugs (14 (58%) vs. 2 (17%); *p* = 0.01), hypersensitivity reactions to food (11 (46%) vs. 0 (0%); *p* < 0.01), and a history of previous positive allergic tests (67% vs. 0%; *p* < 0.001) than those in the control group (Table 1). No difference was found in the overall allergic comorbidity (14 (58%) vs. 4 (33%); *p* = 0.15). Anti-histaminic drugs were used by a higher number of subjects in the study group (*p* = 0.005), whereas no difference was observed for other antiallergic medications. Among the 24 subjects in the study group, 1 subject (4%) reported a previous suspected hypersensitivity reactions to a previous dose of a COVID-19 vaccine, and 12 subjects (50%) were taking antiallergic medications. No difference was found in non-allergic comorbidities between the two groups. None of the participants developed hypersensitivity reactions after vaccine administration. Baseline characteristics of participants are reported in Table 1.

### 3.2. Outcomes

The mean RR variability was lower in patients from the study than those in the control group: 687 ms (645–759) vs. 821 ms (759–902); *p* = 0.02, respectively. SDNN was lower in the study group than in the control group 32 ms (23–36) vs. 50 ms (43–54); *p* < 0.01, respectively. There was no correlation between the age of subjects in the study group and the SDNN values (R^2^ = 0.002). Results of HRV time-domain and frequency-domain measurements are reported in Table 2.

## 4. Discussion

The main finding of our study is that subjects with a severe allergic background have low SDNN and RR values; therefore, individuals with a history of severe allergic reactions seem to have a predominant sympathetic nervous system activation that synchronizes beat-to-beat intervals. We reported a significant prevalence of females in the study group. Additionally, subjects of the study group were significantly younger, and no correlation was found between age and SDNN. 

To the best of our knowledge, this is the first study that intended to evaluate the HRV parameters in subjects with a severe allergic background undergoing COVID-19 vaccination. Alhumaid et al. reported that risk factors for the development of anaphylactic and non-anaphylactic reactions were being female and having a history of atopy [33]. This finding is consistent with our observation; however, Alhumaid et al. found only anamnestic data to detect the risk of allergic reaction to vaccination. Previously, we reported that people with a history of severe allergic reactions undergoing COVID-19 vaccination have a higher level of anxiety compared to individuals with a history of mild allergic reactions [34]. Interestingly, a reduced HRV was historically associated with anxiety disorders [35].

Previous studies evaluated HRV in allergic diseases. Yokusoglu M et al. observed that SDNN was increased in patients with allergic rhinitis [36]. Furthermore, several studies reported the same finding in asthmatic patients [37,38,39], which suggests there is likely a prevalent activation of vagal tone driving pathophysiological pathways. Although the precise pathogenesis of asthma is still debated, one of the proposed underlying mechanisms is an imbalance in the autonomic nervous system [40]. Parasympathetic activity, enhanced cholinergic activation, and vagal tone stimulate bronchoconstriction. Conversely, we reported an increase in sympathetic component activation. This inconsistency may be explained in part by the different natural courses of the analyzed conditions; indeed, the autonomic nervous system enhances excitatory pathways (cholinergic, α-adrenergic, and excitatory non-adrenergic, non-cholinergic-NANC mechanisms) or reduces inhibitory pathways (β-adrenergic and inhibitory NANC mechanisms) [41]. Specifically, norepinephrine inhibits IL-2, interferon, and IL-12 through adrenoceptors and stimulates IL-6 and IL-10 production [42]. Above all, norepinephrine enhances IL-4 production and stimulates IgE production. Furthermore, mast cells have anatomic proximity with postganglionic sympathetic nerve fibers that release norepinephrine [43]. These may all be potential mechanisms underlying hypersensitivity reactions’ development.

There is only one study that reported data on the assessment of HRV parameters in allergic subjects. Consistent with our data, Kazuma et al. reported a reduction in HRV in subjects with a history of hypersensitivity reactions [44]; this suggests that those individuals who experienced hypersensitivity IgE-mediated reactions may have a reduction in HRV, a component of the overactivity of the sympathetic nervous system. Individuals who experienced non-IgE-mediated reactions may be characterized by an overactivity of the parasympathetic system. 

We did not find a correlation between the SDNN and the age of participants. The SDNN has been found to be independent from gender and inversely correlated to age [45]. These results suggest that the differences in baseline characteristics did not influence our results. 

In the study group, participants were taking antiallergic medications (antihistaminic drugs, corticosteroids, anti-leukotrienes drugs) due to their chronic history of allergy. Interactions between the use of antiallergic medications and the HRV parameters were never reported, and no correlation was found between the administration of antiallergic drugs and the HRV measurement [44,46].

The present study has several theoretical and practical implications. First, results from our study suggest the autonomic nervous system might play a key role in allergic conditions. Second, health care professionals and allergists should consider the risk of hypersensitivity reactions could have clinically relevant implications. Indeed, studies have described how a low HRV is associated to a high risk of sudden cardiac death, cardiac arrhythmias, and heart failure [21]; thus, these conditions should be considered especially in a circumstance as stressful as receiving a vaccination.

The need to characterize subjects with a severe allergic background is still debated. Indeed, an objective complementary tool to help in stratification of these subjects is not yet available. The association between low HRV with a severe allergic background could create a way to find a complementary diagnostic tool. In particular, for those subjects who may experience immune and non-immune-mediated adverse events while undergoing vaccination, this may allow safer, large-scale administrations of vaccines and reduce the risk of misdiagnosis. Conditions considered possible risk factors for hypersensitivity reactions to vaccines are a history of uncontrolled asthma or chronic spontaneous urticaria, anaphylaxis, and mast cell disorders with increased tryptase levels [47]. Subjects with a clinical history of previous suspected hypersensitivity reactions to vaccine excipients (such as polyethylene glycol or polysorbate) should undergo an allergy test with these agents before receiving vaccination to exclude potential sensitization and receive vaccination safely. At the end of the allergy workup, subjects considered at a higher risk of hypersensitivity reactions should be referred to exclusive vaccination sessions where they would be supervised by health care personnel trained to address potential moderate to severe allergic reactions [48]. Since a reliable diagnostic tool is not yet available for subjects suspected of being at risk of hypersensitivity reactions, decisions are taken on the basis on anamnestic data only, thus leading to a potential risk of patient mis-stratification. Consequently, some patients may be mistakenly advised to not to be vaccinated, giving them a high risk of contracting severe forms of COVID-19 in case of infection. In addition, further allergic investigations performed on this population may result in unnecessary costs for the health care system. On the other hand, underestimated risks of developing hypersensitivity reactions may lead to unexpected allergic reactions during vaccination. Although the number of patients included in the present study was small, the results are encouraging in supporting a potential future use of HRV as an additional tool to characterize subjects with a relevant history of severe allergic reactions when undergoing vaccination. 

Although our findings should be interpreted with caution, this study has several strengths. We had the possibility of studying a population (those with a relevant past history of severe allergic reactions undergoing vaccination) that is usually difficult to identify; indeed, this was made feasible because of the magnitude of the global pandemic and the Italian strategy to render COVID-19 vaccination mandatory to identify the populations at high risk before vaccination. Second, as opposed to previous studies, we included a control group in our study. 

However, we acknowledge some limitations: first, our study population was comprised of individuals deemed to be at high risk of hypersensitivity reactions to vaccination by traditional risk factors and not those who had developed allergic reactions after a previous vaccine administration. Indeed, none of the participants developed hypersensitivity reactions after vaccination. Second, it is still not clear if sympathetic hyperactivity can increase the severity of systemic hypersensitivity reactions or if an allergic status causes a dysregulation of the autonomic nervous system by itself. Finally, a limited number of subjects underwent vaccination in the later stages of vaccination campaign (up until 1 February 2022, 46.1 million citizens were fully vaccinated in Italy. On 31 January 2023, 47.9 million citizens were fully vaccinated); therefore, less than 2% of citizens in Italy underwent vaccination in the last year. In our single-center study, we enrolled subjects undergoing vaccination during the study period in our center. Considering this limitation, we decided not to perform a baseline-matched case–control analysis. As a result, some baseline characteristics were significantly different between the individuals of the study and the control group (e.g., age, sex). We cannot exclude that baseline differences between the study group and the control group may have influenced our results. Individuals in the study group were younger and more frequently female. Notably, being female is a known risk factor for the development of anaphylactic and non-anaphylactic reactions, which could explain the higher number of female subjects in the study group than in the control group of the present study; however, in previous studies, people who were female and of a younger age were associated with high HRV in contrast to our findings [45]. It is possible to hypothesize that these baseline characteristics did not influence our results, but we cannot completely exclude them. 

In addition, HRV values in the study group were also low when compared with the expected values in the general population. Furthermore, the control group had HRV values consistent with those expected in the general population. 

Another source of uncertainty is that other factors may have had an influence on HRV measurements. For instance, subjects of the study group used antiallergic medications more frequently than the control group. Thus far, there are no available data suggesting a possible interaction between antiallergic drugs with HRV parameters [44,46]; however, the number of studies in this field is so limited that any conclusion is currently impossible.

In addition, it was not possible to perform subgroup analyses due to the present study’s small sample size. Further studies should be performed to investigate the potential influencing role of the general comorbidities or the type of allergy (e.g., food, drug, respiratory) on the HRV. The autonomic nervous system activity is specific to each allergic phenotype, so further research is required to characterize the sympathetic and parasympathetic system activation profiles in different subpopulations of allergic subjects. Future larger cohort studies with a propensity score matched to the current topic are, therefore, recommended. Furthermore, the present results should be validated in a different setting. Further studies are also needed to increase the knowledge about the role of the autonomic nervous system in allergic conditions. Despite our promising results, additional larger studies will be needed to confirm that HRV can be exploited as a complementary tool to characterize individuals with a history of severe allergic reactions undergoing vaccination against SARS-CoV2, or possibly also against other pathogens, and to minimize the risk of misdiagnosis. 

## 5. Conclusions

We demonstrated that individuals with a relevant allergic history have an unbalanced activity of autonomic nervous system in our cohort. Moreover, individuals with a severe allergic background have low HRV parameters, suggesting an overactivity of the sympathetic nervous system. Further larger cohort studies should be performed to characterize subjects with a severe allergic background and validate these results in another setting. Immunologists should consider this association for the optimal management of their allergy patients.

## Figures and Tables

**Figure 1 vaccines-11-00567-f001:**
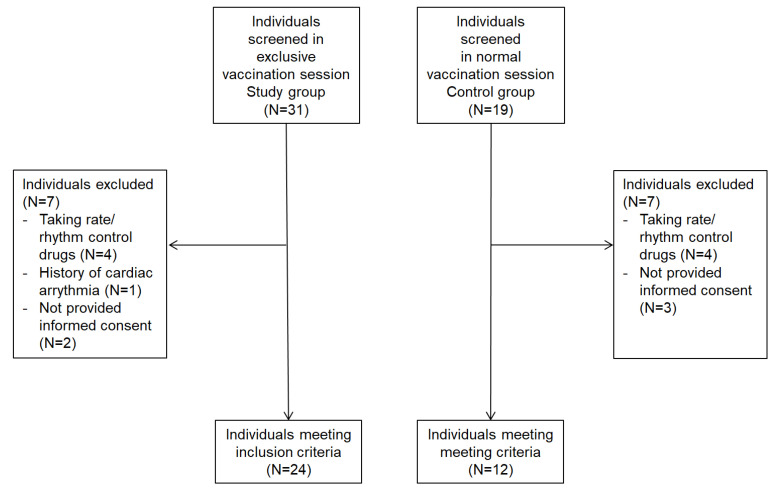
Flowchart illustrating the inclusion/exclusion of individuals in the study.

**Table 1 vaccines-11-00567-t001:** Baseline characteristics of participants.

	Study Group (N 24)	Control Group (N 12)	*p*-Value (<0.05)
Male sex (%)	1 (4%)	3 (25%)	0.02
Age (years), median (interquartile)	47 (34.5–53.3)	59.5(53–66.8)	0.008
Other medical condition	12 (50%)	6 (50%)	0.99
Previous COVID-19 vaccination dose(s)			
0	6 (25%)	1 (8%)	0.2
≥I	18 (75%)	11 (92%)	0.2
Allergic reaction to COVID-19 vaccination	1 (4%)	0 (0%)	0.5
Drugs allergy	14 (58%)	2 (17%)	0.01
Food allergy	11 (46%)	0 (0%)	0.005
Patients with at least one allergic comorbidity, n (%)	14 (58%)	4 (33%)	0.15
Rhinitis	11 (46%)	2 (17%)	0.09
Asthma	5 (21%)	0 (0%)	0.09
Atopic dermatitis	3 (13%)	0 (0%)	0.2
Chronic itch	4 (17%)	0 (0%)	0.1
Contact dermatitis	5 (21%)	1 (8%)	0.3
Chronic urticaria	1 (4%)	0 (0%)	0.5
Positivity to allergic test	16 (67%)	0 (0%)	<0.001
Antiallergic therapies			
Antihistaminic drugs	11(46%)	0 (0%)	0.005
Bronchodilators	4 (17%)	1 (8%)	0.5
Montelukast	2 (8%)	0 (0%)	0.3
Steroid	0 (0%)	0 (0%)	-

**Table 2 vaccines-11-00567-t002:** Heart rate variability parameters in participants.

	Study Group (N 24)	Control Group (N 12)	*p*-Value (<0.05)
RR (ms) median (interquartile)	687(645–759)	821(759–902)	0.02
SDNN (ms) median (interquartile)	32 (23–36)	50 (43–54)	<0.01

## Data Availability

Data presented in this study are available on request from the corresponding author. Data are not publicity available due to privacy.

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
