# Peer review of "Heart Rate Variability in Subjects with Severe Allergic Background Undergoing COVID-19 Vaccination"

_vaccines, 2023, doi:10.3390/vaccines11030567_

Round 1
Reviewer 1 Report
Abstract:
#1: The below version of the abstract can be adopted: I have made some changes.
Anti-Severe Acute Respiratory Syndrome Coronavirus 2 (SARS-CoV-2) vaccination is the world's most important strategy for stopping the pandemic. Vaccination challenges the immune response and can be complicated by hypersensitivity reactions. The autonomic nervous system can modulate the inflammatory immune response, therefore constituting a potential marker to characterize individuals at high risk of hypersensitivity reactions. Autonomic nervous system functionality was assessed through measurement of the heart rate variability (HRV) in subjects with a history of severe allergic reactions and 12 control subjects. HRV parameters included the mean electrocardiograph RR interval and the standard deviation of all normal R-R intervals (SDNN). All measurements were performed immediately before the anti-SARS-CoV-2 vaccination. The median RR variability was lower in the study than in the control group: 687 ms (645-759) vs. 821 ms (759–902); p=0.02. The standard deviation of RR intervals (SDNN) was lower in the study group than in the control group: 32 ms (23–36) vs. 50 ms (43-55); p<0.01. No correlation was found between age and the standard deviation of RR intervals (SDNN). Autonomic nervous system activity is unbalanced in people with a severe allergic background.
#2: The sample size needs more justification
Author Response
Response to Reviewer 1 Comments
Point 1: The below version of the abstract can be adopted: I have made some changes.
Anti-Severe Acute Respiratory Syndrome Coronavirus 2 (SARS-CoV-2) vaccination is the world's most important strategy for stopping the pandemic. Vaccination challenges the immune response and can be complicated by hypersensitivity reactions. The autonomic nervous system can modulate the inflammatory immune response, therefore constituting a potential marker to characterize individuals at high risk of hypersensitivity reactions. Autonomic nervous system functionality was assessed through measurement of the heart rate variability (HRV) in subjects with a history of severe allergic reactions and 12 control subjects. HRV parameters included the mean electrocardiograph RR interval and the standard deviation of all normal R-R intervals (SDNN). All measurements were performed immediately before the anti-SARS-CoV-2 vaccination. The median RR variability was lower in the study than in the control group: 687 ms (645-759) vs. 821 ms (759–902); p=0.02. The standard deviation of RR intervals (SDNN) was lower in the study group than in the control group: 32 ms (23–36) vs. 50 ms (43-55); p<0.01. No correlation was found between age and the standard deviation of RR intervals (SDNN). Autonomic nervous system activity is unbalanced in people with a severe allergic background.
Response 1: We thank the reviewer for the suggestion. The manuscript has been amended accordingly.
Point 2: The sample size needs more justification
Response 2: We thank the reviewer for the possibility to clarify this point. We now better explained this point in the “Sample size” section providing also references which now reads:
“Available data in literature suggests a normal value of the SDNN of 50±10ms [16]. The SDNN < 50ms is associated with an overactivity of the sympathetic nervous sytem. In the latest meta-analysis on the association of inflammation and the HRV, they found that the SNDD was the strongest value negatively correlated with markers of inflammation [24]. Previous studies report that values close to 40ms are associated with clinical manifestations that typically result from an overactivation of sympathetic nervous system [25-28].
Therefore, we identified a reduction of the SDNN to 40ms to be significantly relevant. A sample-size calculation based on Pearson's Chi-square test with a two-sided alpha error of 0.05 and 80% power with an enrolled ratio of two suggested a sample size of 24 participants in study group and 12 participants in control group using the continuity correction, resulting in a total study population of 36 patients.”
Reviewer 2 Report
This study checked the heart rate variability in subjects with severe allergic background before Covid-19 vaccination. Studies have shown that HRV is lower in people suffering from diseases such as diabetes, heart diseases, lung diseases, renal diseases, anxiety disorders, panic attacks, posttraumatic stress disorders, epilepsy, anorexia, and other psychiatric diseases. Therefore, although the HRV of subjects with severe allergic background is lower than control group, the HRV is not a good indicator to characterize individuals with a history of severe allergic reactions.
Author Response
Response to Reviewer 2 Comments
Point 1: This study checked the heart rate variability in subjects with severe allergic background before Covid-19 vaccination. Studies have shown that HRV is lower in people suffering from diseases such as diabetes, heart diseases, lung diseases, renal diseases, anxiety disorders, panic attacks, posttraumatic stress disorders, epilepsy, anorexia, and other psychiatric diseases. Therefore, although the HRV of subjects with severe allergic background is lower than control group, the HRV is not a good indicator to characterize individuals with a history of severe allergic reactions.
Response 1: We thank the reviewer for the careful reading of the manuscript and for his wise observation. We extensively improved discussion where we removed ambigouos sentences on this point.
The sentence:
“To the best of our knowledge, this is the first study that intend to characterize subjects with a severe allergic background undergoing COVID-19 vaccination”
Became:
“To the best of our knowledge, this is the first study that intend to evaluate the HRV parameters in subjects with a severe allergic background undergoing COVID-19 vaccination”
The sentence:
“HRV could represent a potential complementary tool that characterizes severe allergic patients undergoing exclusive vaccination session which may experience adverse events both immune and non-immune mediated, thus allowing a safer large scale administrations of vaccines and reducing the risk of misdiagnosis.”
Became:
“The association between low HRV with a severe allergic background could open the way to find a complementary diagnostic tool. Especially for those subjects undergoing vaccination which may experience adverse events both immune and non-immune mediated, this may allow a safer large scale administrations of vaccines and reduce the risk of misdiagnosis.”
And conclusion:
“We showed that in our cohort individuals with a relevant allergic history have an unbalanced activity of autonomic nervous system. HRV measurement could represent a potential promising tool to characterize subjects with a severe allergic background undergoing vaccination in exclusive sessions of vaccination.”
Became:
“We showed that in our cohort individuals with a relevant allergic history have an unbalanced activity of autonomic nervous system. Individuals with a severe allergic background have low HRV parameters suggesting an overactivity of sympathetic nervous system.”
Reviewer 3 Report
This manuscript by Bernadette and others examines heart rates in allergic history and non-allergic controls. The data and manuscript are very simple and easy to follow.
Abbreviations must be spelled out only at the first appearance. The authors spell out SDNN 3 times in the Abstract.
Table 1: ages in the study group are far younger than the control, which may affect the results. The authors must have used age-matched control.
Table 1: 46% of study group participants take antihistaminic drugs, while the number is 0% in the control group. Can the authors make sure antihistaminic drugs did not affect the outcomes?
Author Response
Response to Reviewer 3 Comments
Point 1: This manuscript by Bernadette and others examines heart rates in allergic history and non-allergic controls. The data and manuscript are very simple and easy to follow.
Response 1: We thank the reviewer for appreciating our manuscript and for her/his positive comments.
Point 2: Abbreviations must be spelled out only at the first appearance. The authors spell out SDNN 3 times in the Abstract.
Response 2: We thank the reviewer for noticing this error. We corrected the manuscript as suggested.
Point 3: Table 1: ages in the study group are far younger than the control, which may affect the results. The authors must have used age-matched control.
Response 3: We thank the reviewer for sharing this concern. We updated the manuscript with reasons of these discrepancies. We performed this single-centre study in last stages of vaccination campaign. Just a 2% of citiziens underwent vaccination in Italy, and few in our centre. Therefore, enrolled all those subjects undergoing vaccination in our centre. This did not allow us to match age between groups.
We extensively explained our reasoinings and now you can read in “Methods”:
“Due to the limited number of subjects undergoing vaccination in the later stages of vaccination campaign, subjects were sequentially selected. Therefore, we decided to select for the enrollment the first 24 subjects of the study group and the first 12 subjects of the control group meeting inclusion criteria.”
In “Discussion”:
“We did not find correlation between the SDNN and the age of participants. The SDNN is independent from gender and inverserly correlated to age [45]. These results suggest that the differences in baseline characteristics did not influence our outcome.”
And in “Limitations”:
“Finally, a limited number of subjects underwent vaccination in the later stages of vaccination campaign (Up to 1st February 2022, 46,1 millions citiziens were fully vaccinated in Italy. On 31st January 2023, 47,9 millions citiziens were fully vaccinated). Therefore, less than 2% of citizien in Italy underwent vaccination in the last year. In our single-centre study, we enrolled all those subjects undergoing vaccination in that study period in our centre. Considering this limitation, we decided not to perform a baseline-matched case control analysis. As a results, some baseline characteristics were significantly different between the individuals of the study and of the control group (e.g. age, sex).”
Point 4: Table 1: 46% of study group participants take antihistaminic drugs, while the number is 0% in the control group. Can the authors make sure antihistaminic drugs did not affect the outcomes?
Response 4: We thank the reviewer for the opporturnity to clarify this point. There is not association bewteen the administration of antiallergic drugs and HRV measurement. The only drugs which can influence HRV parameters are those considered as exclusion criteria for our study.
We better clarified this point in “Discussion” providing also references:
“In the study group participants were taking antiallergic medications (antihistaminic drugs, corticosteroids, anti-leukotrienes drugs) due to their chronic history of allergy. Interactions between the use of antiallergic medications and the HRV paramenters were never reported and no correlation was found between the administration of antiallergic drugs and the HRV measurement [44, 46].”
And in “Methods”:
“Subjects of the study group have peculiar characteristics related to the allergic history (e.g. antiallergic medications, positivity to allergic test); therefore, it was not possible to match individuals of the study group with those of the control group.”
Reviewer 4 Report
I am really , but there is no way this study can be accepted for publication in a reputable journal.
The main problem with this study is that control for confounders has been totally neglected, nor even mentioned in the study limitations.
HRV can be caused by a number fo factors including heart, renal and psychological factors and the authors considered only allergic medical history.
Furthermore, baseline characteristics of case an controls ( table 1) are quite different. For instance controls are on average 13 years older than cases and the vast majority of subjects were females. Any of those baseline discrepancies could easily explain the difference in study outcome.
The only factor considerred in the analysis is difference in HRV by study sujects (allergic?) and controls (unalelrgic).
There is no mention how study subjects and controls were selected. Were they all allergic patients? or controls were not allergic? this is not explained at all in methods.
Sample size calculation was not enough described. What are the assumptions behind these numbers ? In any case 36 patients (24 vs 12) is a limited sample to draw similar conlcusions. Not to mention that typically the number of controls overcome the number of cases.
Overall, this study makes no sense really,
I am sorry to be franked,
Cordially.
Author Response
Response to Reviewer 4 Comments
Point 1: I am really , but there is no way this study can be accepted for publication in a reputable journal.
Response 1: We thank the reviewer for reviewing our manuscript.
Point 2: The main problem with this study is that control for confounders has been totally neglected, nor even mentioned in the study limitations.
Response 2: We understand the point of view of the reviewer. Unfortunately, in observational studies, bias of confounding is not always avoidable. Indeed, we observed the association between a low HRV in subjects with a history of severe allergic reactions. Normal values of HRV were already described in general population. What we did to strengthen our results was to compare subjects of the study group with the control group which was selected according to the expected standards for this type of study. Indeed, the control group was defined as a group of subjects undergoing Covid19 vaccination, and with a low risk of developing hypersensibility reactions to vaccination. We performed this single-centre study in the last stages of vaccination campaign. Only 2% of citiziens underwent vaccination in Italy, and few in our centre. Therefore, we enrolled all those subjects undergoing vaccination in our centre. This did not allow us to match baseline characteristics between groups.
Thanks to the reviewer we extensively improved the manuscript, and now you can read in “Methods”:
“Due to the limited number of subjects undergoing vaccination in the later stages of vaccination campaign, subjects were sequentially selected. Therefore, we decided to select for the enrollment the first 24 subjects of the study group and the first 12 subjects of the control group meeting inclusion criteria.”
And in “Limitations”:
“Finally, a limited number of subjects underwent vaccination in the later stages of vaccination campaign (Up to 1st February 2022, 46,1 millions citiziens were fully vaccinated in Italy. On 31st January 2023, 47,9 millions citiziens were fully vaccinated). Therefore, less than 2% of citizien in Italy underwent vaccination in the last year. In our single-centre study, we enrolled all those subjects undergoing vaccination in that study period in our centre. Considering this limitation, we decided not to perform a baseline-matched case control analysis. As a results, some baseline characteristics were significantly different between the individuals of the study and of the control group (e.g. age, sex).”
Point 3: HRV can be caused by a number fo factors including heart, renal and psychological factors and the authors considered only allergic medical history
Response 3: We thank the reviewer for this observation. We decided to study and assess HRV paramenters only related to the status of allergic background. No difference was found in overall non–allergic comorbidities and we reported these data in results.
Point 4: Furthermore, baseline characteristics of case an controls ( table 1) are quite different. For instance controls are on average 13 years older than cases and the vast majority of subjects were females. Any of those baseline discrepancies could easily explain the difference in study outcome.
Response 4: We thank the reviewer to giving us the opportunity to better explain this point. In previous response we explain the reasoning of this discrepancy, and we consequently improved the manuscript following the reviewer suggestion.
We also reported in “Discussion”:
“We did not find correlation between the SDNN and the age of participants. The SDNN is independent from gender and inverserly correlated to age [45]. These results suggest that the differences in baseline characteristics did not influence our outcome”
Point 5: The only factor considerred in the analysis is difference in HRV by study sujects (allergic?) and controls (unalelrgic).
There is no mention how study subjects and controls were selected. Were they all allergic patients? or controls were not allergic? this is not explained at all in methods.
Response 5: We thank the reviewer for asking this clarification about selection of participants. We updated the “Methods” which now reads:
“Subjects undergoing vaccination and older than 18 years were eligible for the study. Patients with cardiac arrhythmias, or taking rate/rhythm control drugs, or who did not provide written informed consent to participate were excluded. Subjects with a severe allergic background were previously identified by an allergist independently from the study. Those individuals considered at high risk of hypersensitivity reactions are referred to exclusive vaccination sessions. Subject with a history of severe hypersensitivity reactions were identified on the basis of anamnestic data through the administration of a questionnaire assessing history of atopy, antiallergic medication, previous allergic reactions to drugs, food, or reactions to previous doses of Covid19 vaccine. The control group was selected according to the expected standards for this type of study. Indeed, the control group was defined as a group of subjects undergoing Covid19 vaccination with a low risk of developing hypersensibility reactions to vaccination. The low risk of developing hypersensibility reactions to vaccination was defined from the allergits based on the clinical history.”
Point 6: Sample size calculation was not enough described. What are the assumptions behind these numbers ? In any case 36 patients (24 vs 12) is a limited sample to draw similar conlcusions. Not to mention that typically the number of controls overcome the number of cases.
Response 6: We thank the reviewer for the possibility to clarify this point. We now better explained this point in the Sample size section providing also references which now reads:
“Available data in literature suggests a normal value of the SDNN of 50±10ms [16]. The SDNN < 50ms is associated with an overactivity of the sympathetic nervous sytem. In the latest meta-analysis on the association of inflammation and the HRV, they found that the SNDD was the strongest value negatively correlated with markers of inflammation [24]. Previous studies report that values close to 40ms are associated with clinical manifestations that typically result from an overactivation of sympathetic nervous system [25-28].
Therefore, we identified a reduction of the SDNN to 40ms to be significantly relevant. A sample-size calculation based on Pearson's Chi-square test with a two-sided alpha error of 0.05 and 80% power with an enrolled ratio of two suggested a sample size of 24 participants in study group and 12 participants in control group using the continuity correction, resulting in a total study population of 36 patients.”

Round 2
Reviewer 2 Report
The new version improved the statements and avoid the insufficient conclusions.
Author Response
Response to Reviewer 2 Comments
Point 1: The new version improved the statements and avoid the insufficient conclusions.
Response 1: We thank the reviewer to appreciate the new version of the manuscript.
Reviewer 3 Report
The authors have modified the manuscript, almost as I expected.
Author Response
Response to Reviewer 3 Comments
Point 1: The authors have modified the manuscript, almost as I expected.
Response 1: We thank the reviewer for appreciating the new version of the manuscript.
Reviewer 4 Report
This study is not enough sound.
Againg HRV can be sustained by a number of factors not considered in the analysis. Moreover, there are significant baseline differebce between cases and controls which might explain HRV.
Author Response
Response to Reviewer 4 Comments
Point 1: This study is not enough sound.
Againg HRV can be sustained by a number of factors not considered in the analysis. Moreover, there are significant baseline differebce between cases and controls which might explain HRV.
Response 1: We understand the reviewer point of view. We deeply improved limitation according to the reviewer concerns and now reads:
“We can not exclude that baseline differences between the study group and the control group may influenced our final results. Individuals in the study group were younger and more frequently females. Notably, female sex is a known risk factors for the development of anaphylactic and non-anaphylactic reactions. This could explain an higher number of female subjects in the study group than in the control group. However, in previous studies female sex and younger age were associated with high HRV in contrast with our findings [45]. It is possible to hypothesise that these baseline characteristics did not influence our results, but we can not totally exclude it.
In addition, HRV values in the study group were low also when compared with the expected values in general population. Furthermore, the control group had HRV values consistent with those expected in the general population.
Another source of uncertainty is that other factors may have an influence on HRV measurement. For instance, subjects of the study group used more frequently an-ti-allergic medications. So far, there are not available data suggesting a possible interaction of anti-allergic drugs with HRV parameters [44, 46]. However, the number of studies in this field are so limited that any conclusion is impossible.
In addition, it was not possible to perform subgroup analyses due to the small sample size. Further studies should be performed to investigate the potential influencing role of the general comorbidities or the type of allergy (e.g., food, drug, respiratory) on the HRV. Indeed, the autonomic nervous system activity is specific to each allergic phenotype. Further work is required to characterize sympathetic and parasympathetic system activation profiles in different subpopulations of allergic subjects. Future larger cohort studies with a propensity score matched on the current topic are therefore recommended. Furthermore, the present results need to be validated in different setting.”
Round 3
Reviewer 4 Report
I am afraid but I cannot see any reason why to insist to consider a similar study for publication.
In addition to the already mentioned epidemiological weakness:
- limited sample size (24 cases vs 12 controls);
- unbalanced distribution of baseline variables;
- Zero control for confounders
HRV is a stress indicator, sustained by several factors not considered in the analysis. It is expected that allergic indivividuals may have different HRV for fear of being immunized with a new vaccine, but claiming that HRV can be use to predict the risk of allergic reaction is simply misleading.
Not a surprise that there is only one published study on this topic, since this hypothesis makes no sense.
Unless proven with sound scietintific research, which is not the case with this biased study.
Author Response
Point 1: I am afraid but I cannot see any reason why to insist to consider a similar study for publication.
In addition to the already mentioned epidemiological weakness:
- limited sample size (24 cases vs 12 controls);
- unbalanced distribution of baseline variables;
- Zero control for confounders
HRV is a stress indicator, sustained by several factors not considered in the analysis. It is expected that allergic individuals may have different HRV for fear of being immunized with a new vaccine, but claiming that HRV can be use to predict the risk of allergic reaction is simply misleading.
Not a surprise that there is only one published study on this topic, since this hypothesis makes no sense.
Unless proven with sound scietintific research, which is not the case with this biased study.
Response 1:
We thanks the reviewer for his effort in reading the manuscript and we understand his point of view. We extensively reported limitations of the study in the discussion section and methods at each round of revision.
Considering the setting and the special population of individuals we can not extend sample size.
We deeply justified in the introduction and discussion the reason of the association between HRV and allergic patients. We believe that biological mechanisms of hypersensitivity reaction is a growing field in research as we mention in discussion.